# BCL11A interacts with SOX2 to control the expression of epigenetic regulators in lung squamous carcinoma

Kyren A. Lazarus[1,2], Fazal Hadi [1,2], Elisabetta Zambon[1,2], Karsten Bach[1,2], Maria-Francesca Santolla [1,3], Julie K. Watson[4], Lucia L. Correia[5], Madhumita Das[6], Rosemary Ugur[1,2], Sara Pensa[1,2], Lukas Becker[1], Lia S. Campos[7], Graham Ladds[1], Pentao Liu [7], Gerard I. Evan[5], Frank M. McCaughan[5], John Le Quesne[6,8,9], Joo-Hyeon Lee [4], Dinis Calado[10] & Walid T. Khaled [1,2,4]

Patients diagnosed with lung squamous cell carcinoma (LUSC) have limited targeted therapies. We report here the identification and characterisation of *BCL11A*, as a LUSC oncogene. Analysis of cancer genomics datasets revealed *BCL11A* to be upregulated in LUSC but not in lung adenocarcinoma (LUAD). Experimentally we demonstrate that non-physiological levels of *BCL11A* in vitro and in vivo promote squamous-like phenotypes, while its knockdown abolishes xenograft tumour formation. At the molecular level we found that *BCL11A* is transcriptionally regulated by SOX2 and is required for its oncogenic functions. Furthermore, we show that BCL11A and SOX2 regulate the expression of several transcription factors, including *SETD8*. We demonstrate that shRNA-mediated or pharmacological inhibition of SETD8 selectively inhibits LUSC growth. Collectively, our study indicates that BCL11A is integral to LUSC pathology and highlights the disruption of the BCL11A–SOX2 transcriptional programme as a novel candidate for drug development.

[1] Department of Pharmacology, University of Cambridge, Cambridge CB2 1PD, UK. [2] Cambridge Cancer Centre, CB2 0XZ Cambridge, UK. [3] Department of Pharmacy, Health and Nutritional Sciences, University of Calabria, Rende 87036, Italy. [4] WT-MRC Stem Cell Institute, University of Cambridge, Cambridge CB2 0SZ, UK. [5] Department of Biochemistry, University of Cambridge, Cambridge CB2 1GA, UK. [6] MRC Toxicology Unit, Lancaster Road, Leicester LE1 7HB, UK. [7] Wellcome Trust Sanger Institute, Cambridge CB10 1SA, UK. [8] Cancer Research Centre, University of Leicester, Leicester LE2 7LX, UK. [9] University Hospitals Leicester NHS trust, Leicester LE1 5WW, UK. [10] The Francis Crick Institute, London NW1 1AT, UK. Correspondence and requests for materials should be addressed to W.T.K. (email: wtk22@cam.ac.uk)

Lung cancer accounts for the highest rate of cancer-related diagnosis and mortality worldwide[1]. Broadly, there are two major types of lung cancers; small cell lung cancer (SCLC) accounting for 15% of cases and non-small cell lung cancer (NSCLC) accounting for 85% of cases[1]. NSCLC patients have a poor outcome in the clinic with only 15% of patients surviving 5 years or more[2]. At present NSCLC is defined histo-pathologically in the clinic into four broad categories: lung adenocarcinoma (LUAD), lung squamous cell carcinoma (LUSC), large cell carcinoma and undifferentiated NSCLC. LUAD and LUSC are the most common types of NSCLC (90% of cases). LUSC accounts for more than 400,000 deaths worldwide each year[3] and unlike LUAD there are limited targeted therapies available for LUSC. Platinum-based chemotherapy remains the first-line treatment for LUSC and although the recent FDA approval of Necitumumab in combination with platinum-based chemotherapy for metastatic LUSC has shown positive signs, a great deal of work still needs to be done in this field[4].

At the cellular level, LUAD tends to originate from the secretory epithelial cells in the lung while LUSC usually originates from basal cells (BCs) in the main and central airways[2]. At the molecular level LUAD is known to harbour mutations in epidermal growth factor receptor (EGFR), V-Ki-Ras2 Kirsten Rat Sarcoma Viral Oncogene Homologue (KRAS) and Anaplastic Lymphoma Receptor Tyrosine Kinase (ALK), which are also well modelled and studied both in vitro and in vivo (for review see refs. [5,6]). On the other hand, LUSC is less studied but it is known that amplifications of Sex Determining Region Y (SRY)-Box 2 (SOX2) tend to be present in 70–80% of patients[7–10]. We report here the identification and characterisation of the transcriptional regulator, BCL11A as a LUSC oncogene. We demonstrate that along with SOX2, it regulates key epigenetic factors, and that the disruption of one of these factors, SETD8 leads to selective affects in LUSC cells compared to LUAD cells. By disrupting the BCL11A-SOX2 transcriptional programme, our results provide a potential future framework for tackling the unmet clinical need for LUSC patients.

## Results

**BCL11A is a LUSC oncogene.** Recently, a detailed picture of the molecular differences between LUAD and LUSC has been made available through 'The Cancer Genome Atlas' (TCGA)[11,12]. To identify key drivers responsible for the differences between LUAD and LUSC we reanalysed the gene expression data from TCGA and focused on transcriptional regulators in the genome. As reported previously SOX2 was the most amplified gene in LUSC and its expression level was also significantly higher in LUSC vs. LUAD (Fig. 1a and Supplementary Fig. 1a). The second most amplified locus in LUSC patients revealed by TCGA analysis contains the transcription factors BCL11A and REL[11,12]. BCL11A has been shown to be an oncogene in B-cell lymphoma and triple negative breast cancer[13–16].

We found that BCL11A expression levels were also significantly higher in LUSC vs. LUAD (Fig. 1a and Supplementary Fig. 1a). Moreover, the expression of both BCL11A and SOX2 was significantly higher in LUSC but not in LUAD tumour samples compared to patient matched normal samples (Fig. 1b, c and Supplementary Fig. 1b–c) supporting a driver role for these transcription factors in LUSC pathology. In contrast, REL expression was unchanged between LUSC and LUAD (Fig. 1a–c and Supplementary Fig. 1a–c) suggesting that BCL11A amplification is a key driving event in LUSC. This observation is supported by the recent report from the TRACERx (TRAcking Cancer Evolution through therapy (Rx)) study demonstrating the amplification of BCL11A as an early event in LUSC[17].

Furthermore, BCL11A IHC staining on LUAD ($n = 99$) and LUSC ($n = 120$) TMAs revealed little or no staining in 99% of LUAD patients. In contrast, 25% of LUSC patients had moderate staining and 24% of LUSC patients had strong staining, which is in agreement with previous IHC staining of NSCLC tumours[18] (Fig. 1d). This confirms our transcriptomic analyses indicating the specificity of BCL11A in LUSC patients.

To determine if high levels of BCL11A expression are oncogenic in LUSC, we performed shRNA-mediated knockdown (KD) of BCL11A using two independent shRNAs in two LUSC cell lines, LK2 and H520 (Supplementary Fig. 2a and b). We first tested the clonogenic capacity of control or BCL11A-KD cells by seeding them into matrigel for 3D colony formation assays. We found that BCL11A-KD cells had a significant reduction in colony formation capacity (Supplementary Fig. 2c and d). We then injected control or BCL11A-KD cells into immune compromised mice to test for their tumour formation capacity and found a significant reduction in tumour burden from BCL11A-KD cells compared to control cells (Fig. 1e, f). In addition, we found the squamous markers KRT5 and TP63 levels were significantly reduced in BCL11A-KD LUSC cells (Supplementary Fig. 2e–h). However, we found no change in the expression of SOX2 in BCL11A-KD LUSC cells suggesting that BCL11A activity is downstream of SOX2 (Supplementary Fig. 2i and j). Moreover, to test if the role of BCL11A is context dependant, we knocked down BCL11A in a LUAD cell line H1792 and found no change in 3D colony growth indicating specificity at the cellular level (Supplementary Fig. 2k–l).

**BCL11A overexpression leads to thickening of the airways.** To explore the role of BCL11A in lung biology, we utilised a novel Cre-inducible mouse model that allows for the overexpression of BCL11A. Essentially, BCL11A was inserted into the Rosa26 locus with a LoxP-Stop-LoxP (LSL) cassette upstream, under the control of a CAGG promoter, thus preventing the expression of BCL11A unless the LSL is excised by Cre recombinase. To test the effect of BCL11A overexpression on lung morphology, we allowed the BCL11A-overexpression (BCL11A^{ovx}) mice to inhale Adenovirus-Cre (Fig. 2a). Eight months after infection, we analysed the lungs and found signs of airway hyperplasia (Fig. 2b, c) accompanied by aggregates of small hyperchromatic cells with irregular nuclei that represent reserve cell hyperplasia (Fig. 2b, arrows and inset), which are precursors to squamous metaplasia[19]. IHC analysis of the lungs from BCL11A^{ovx} also indicated an increase in positivity for the proliferative marker Ki-67 (Supplementary Fig. 3a) and Sox2 indicating a transition to squamous differentiation (Supplementary Fig. 3b). However, we found little difference in Cc10, Krt5 and Trp63 staining at this stage (Supplementary Fig. 3a and b).

To further investigate the role of BCL11A in LUSC, we employed a 3D organoid culture system for BCs from the trachea, as they have been suggested to be the cell of origin for LUSC[20–22]. BCs from human and mouse lungs have higher expression of BCL11A when compared to the other epithelial compartments indicating its importance in lung biology (Supplementary Fig. 4a and b)[22]. Therefore, we crossed the BCL11A^{ovx} mice to R26-CreERT2 mice, which allowed us to induce Cre recombination by the administration of tamoxifen. In addition, we also used a Bcl11a conditional knockout mouse under the control of R26-CreERT2 (called Bcl11a^{CKO} from this point) to elucidate the importance of BCL11A in tracheosphere organoid formation (Fig. 2a).

We found that in contrast to the hollow organoids normally formed by BCL11A^{OVX} or Bcl11a^{CKO} organoids treated with ethanol in vitro, BCs from BCL11A^{ovx} mice treated with

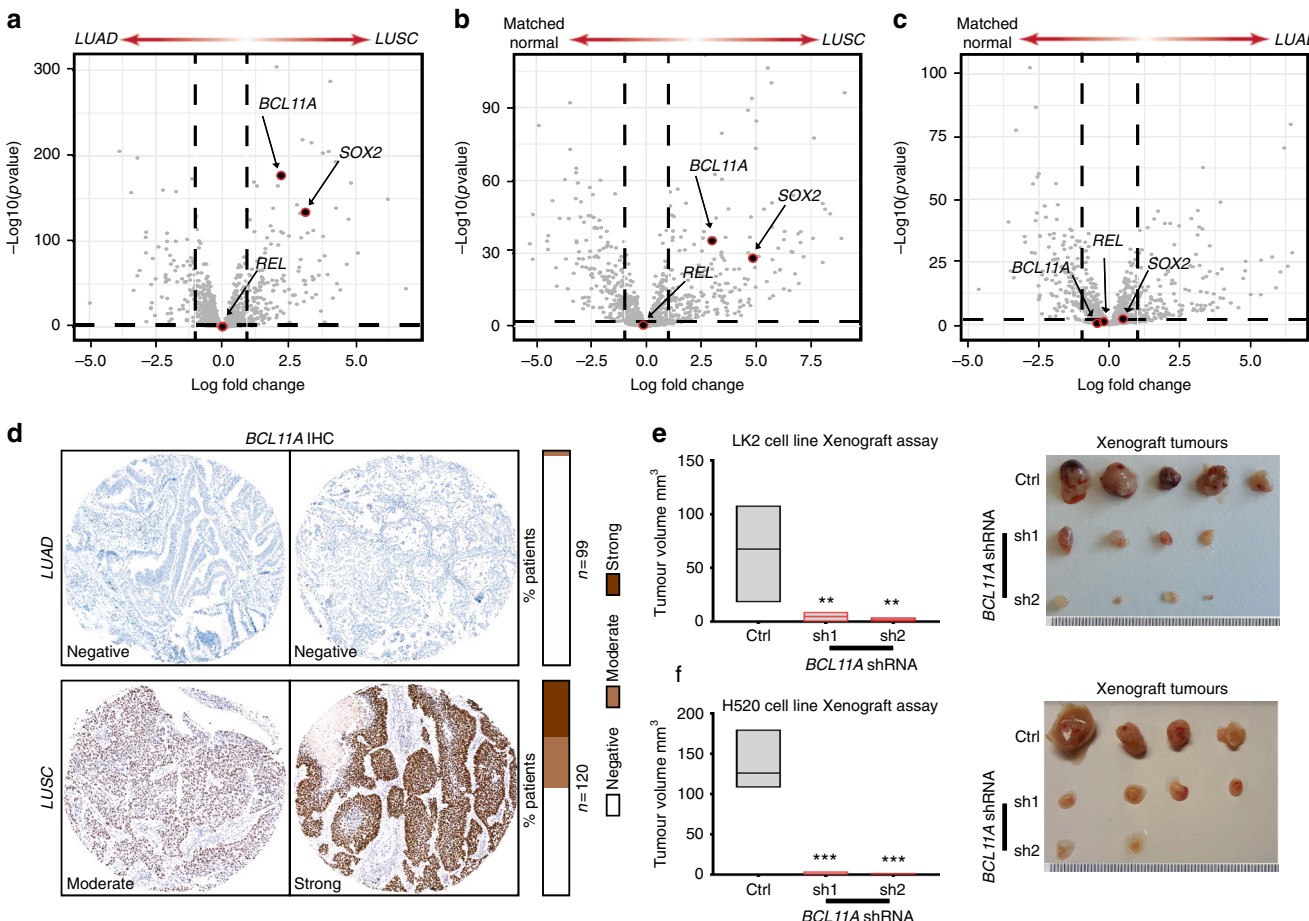

**Fig. 1** *BCL11A* is a lung squamous cell carcinoma (LUSC) oncogene. **a** Volcano plots of The Cancer Genome Atlas (TCGA) RNAseq data[11, 12] indicating that *BCL11A* and *SOX2* are highly expressed in LUSC compared to lung adenocarcinoma (LUAD). The plots show that *REL* is not differentially expressed in LUSC vs. LUAD patients. The x-axis represents $\log_2$ expression fold-change (FC) in LUSC patients vs. LUAD patients and the y-axis represents $-\log_{10}$(pValue). The vertical dashed lines represent FC = 1.0 and the horizontal dashed line represents p-value = 0.01. **b** Volcano plots indicating that *BCL11A* and *SOX2* are differentially expressed in LUSC patients vs. matched normal samples. The plot indicates that *REL* is not differentially expressed in LUSC vs. matched normal samples. **c** Volcano plots indicating that neither BCL11A, SOX2 nor REL are differentially expressed in LUAD patients vs. matched normal. **d** Images and scoring of BCL11A IHC staining on LUAD and LUSC tumours (see Methods for scoring). **e**, **f** Graphs depicting reduction in tumour size observed when shRNA1 or shRNA2 against *BCL11A*-transfected LK2 (**e**) and H520 (**f**) cells are injected subcutaneously into mice compared to control. Five mice per cell line were monitored for 25 days after which tumours were removed and measured. On the right are images showing actual tumours measured. The box plot extends from 25th to 75th percentile and the line represents the median. Data presented as mean ± s.d. One-way ANOVA with post Dunnett test performed, $^*p < 0.05$, $^{**}p < 0.005$, and $^{***}p < 0.001$

tamoxifen in vitro formed solid organoid structures with no hollow lumen suggesting hyper-proliferation and loss of organisation (Fig. 2d). However, BCs from *Bcl11a^cko* mice failed to form any organoid structures suggesting that *Bcl11a* is necessary for organoid formation (Fig. 2d). Quantitative analysis revealed a significant decrease in organoid numbers from *Bcl11a^cko* BCs but no significant difference in *BCL11A^ovx* BCs (Supplementary Fig. 4c and d). Tamoxifen or ethanol-treated organoids from WT mice showed no difference in organoid morphology indicating that the differences are attributed to changes in *BCL11A* expression (Supplementary Fig. 4e)

H&E staining confirmed the hollowness of the organoids from the control mice which was in stark contrast to the filled organoids from the *BCL11A^ovx* mice (Fig. 2e and Supplementary Fig. 5). IF staining revealed that the *BCL11A^ovx* organoids were also positive for Ki-67 (Supplementary Fig. 5), Sox2, Krt5, Trp63 and negative for the luminal marker Krt8 (Fig. 2e and Supplementary Fig. 6) indicating that BCL11A maintains a squamous phenotype.

**BCL11A and SOX2 occupy common loci in LUSC cells.** Given the importance of SOX2 in driving LUSC[8,9] we next investigated if *BCL11A* is regulated by SOX2. To achieve this, we knocked down *SOX2* (SOX2-KD) in two LUSC cell lines using two independent shRNAs (Fig. 3a, b and Supplementary Fig. 7a and b). Similar to *BCL11A* we found that *SOX2-KD* cells had a significantly reduced colony and tumour formation abilities (Supplementary Fig. 7c–f). At the molecular level we found a significant reduction in the expression levels of *BCL11A* and similar to *BCL11A-KD*, reduction in squamous markers *KRT5* and *TP63* in the *SOX2-KD* cells (Fig. 3c, d and Supplementary Fig. 7g–j).

To investigate if BCL11A is required for SOX2-mediated oncogenesis we introduced a doxycycline inducible *BCL11A* overexpression vector into *SOX2-KD* cell lines and found that *BCL11A* overexpression partially rescues the colony and tumour formation abilities of *SOX2-KD* cells (Fig. 3e–h, and Supplementary Fig. 8a–c). These results suggest that BCL11A is partially responsible for SOX2's-mediated transcriptional changes in LUSC cells.

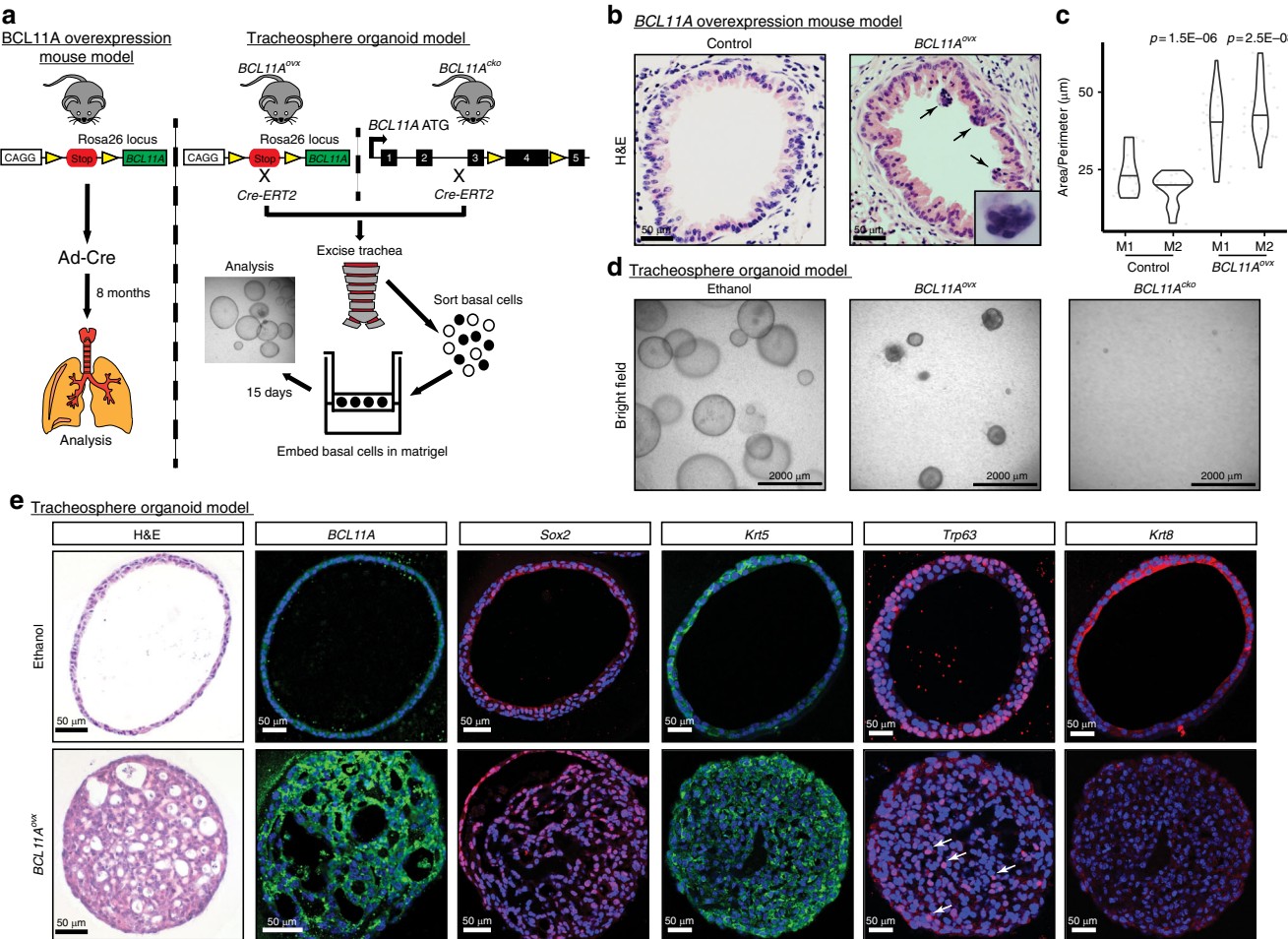

**Fig. 2** BCL11A overexpression leads to thickening of the airways and abnormal organoid formation. **a** Schematic representing strategy to explore the role of BCL11A in vivo and ex vivo. Left Panel: Adenovirus-Cre was nasally administered to BCL11A$^{ovx}$ mice and the lungs were analysed after eight months. Right panel: for the tracheosphere organoid model, basal cells from the trachea of either BCL11A$^{ovx}$ or BCL11A$^{cko}$ mice were FACS sorted, embedded in matrigel and analysed after 15 days. Three independent mice were used for each experiment. **b** Images of airways from control and BCL11A$^{ovx}$. Arrows indicate small hyperchromatic cells with irregular nuclei. **c** Quantificaiton of airway epithelial layer hyperplasia from two control and BCL11A$^{ovx}$ mice. **d** Bright field images of organoids from Bcl11a$^{cko}$ and BCL11A$^{ovx}$ mice treated with vehicle or tamoxifen. **e** Sectioned organoids from BCL11A$^{ovx}$ mice stained with haematoxylin and eosin, BCL11A, Sox2, Krt5, Trp63 and Krt8. Arrows indicate positive staining. Scale bar indicates 50 μm

To understand how BCL11A and SOX2 contribute to the LUSC transcriptional programme we performed BCL11A and SOX2 ChIP-Seq analysis on LK2 cells in the presence or absence of BCL11A (BCL11A-KD) (Fig. 3i). We identified 49,567 peaks for SOX2 and 4294 peaks for BCL11A (Fig. 3j and Supplementary Data 1 and 2). Out of the 4294 BCL11A peaks identified, 3946 were not present in BCL11A-KD cells validating the true nature of BCL11A binding at these regions of the genome.

We then compared regions bound by both BCL11A and SOX2 in LK2 control cells and identified 1114 peaks suggesting that a relationship between these two transcription factors (Fig. 3i, j). Subsequently, we identified the nearest genes to the common peaks (Supplementary Data 3) and performed Gene Ontology (GO) analysis to identify if these common peaks are enriched for specific biological process (Fig. 3j). The top hits from the GO analysis revealed enrichment for transcriptional and epigenetic regulators including SETD8, SKIL, and TBX2 (Fig. 3k, l). These three factors have been reported to have roles in NSCLC. SKIL and TBX2 have been reported to be upregulated in NSCLC[23,24]. SETD8 on the other hand has been indirectly linked to NSCLC as a target of mir-382 in NSCLC[25]. We also found that SOX2 binds the BCL11A locus at multiple sites suggesting a strong direct

regulation at the transcriptional level further supporting the data in Fig. 3c, d–l. BCL11A and SOX2 peaks on SETD8[26,27], SKIL[28], TBX2[29] and BCL11A were validated and confirmed by ChIP-qPCR (Supplementary Fig. 9). The overlap of the ChIP-Seq peaks suggests a direct interaction between BCL11A and SOX2 proteins, which was confirmed in co-immunoprecipitation experiments on LK2 and H520 (Supplementary Fig. 10a, b).

**BCL11A and SOX2 regulate SETD8 gene expression**. To further assess the importance of these three factors we first analysed the expression of SETD8, SKIL and TBX2 in multiple NSCLC cell lines and found that SETD8 and SKIL expression was significantly higher in LUSC cell lines ($n = 5$) compared to LUAD cell lines ($n = 6$) (Fig. 4a, b) which was in correlation with the expression levels of BCL11A and SOX2 (Fig. 4d, e). In contrast, TBX2 levels were not different in LUSC cell lines vs. LUAD cell lines (Fig. 4c). To understand if BCL11A regulates these genes, we sorted basal stem cells (Epcam$^{+ve}$/GSIβ4$^{+ve}$)[30,31] and Epcam$^{+ve}$ cells from trachea excised from tamoxifen-treated BCL11A$^{ovx}$ or WT mice (Fig. 4f, g and k). We found that Setd8 and Skil, but not Tbx2 and Sox2 are upregulated in

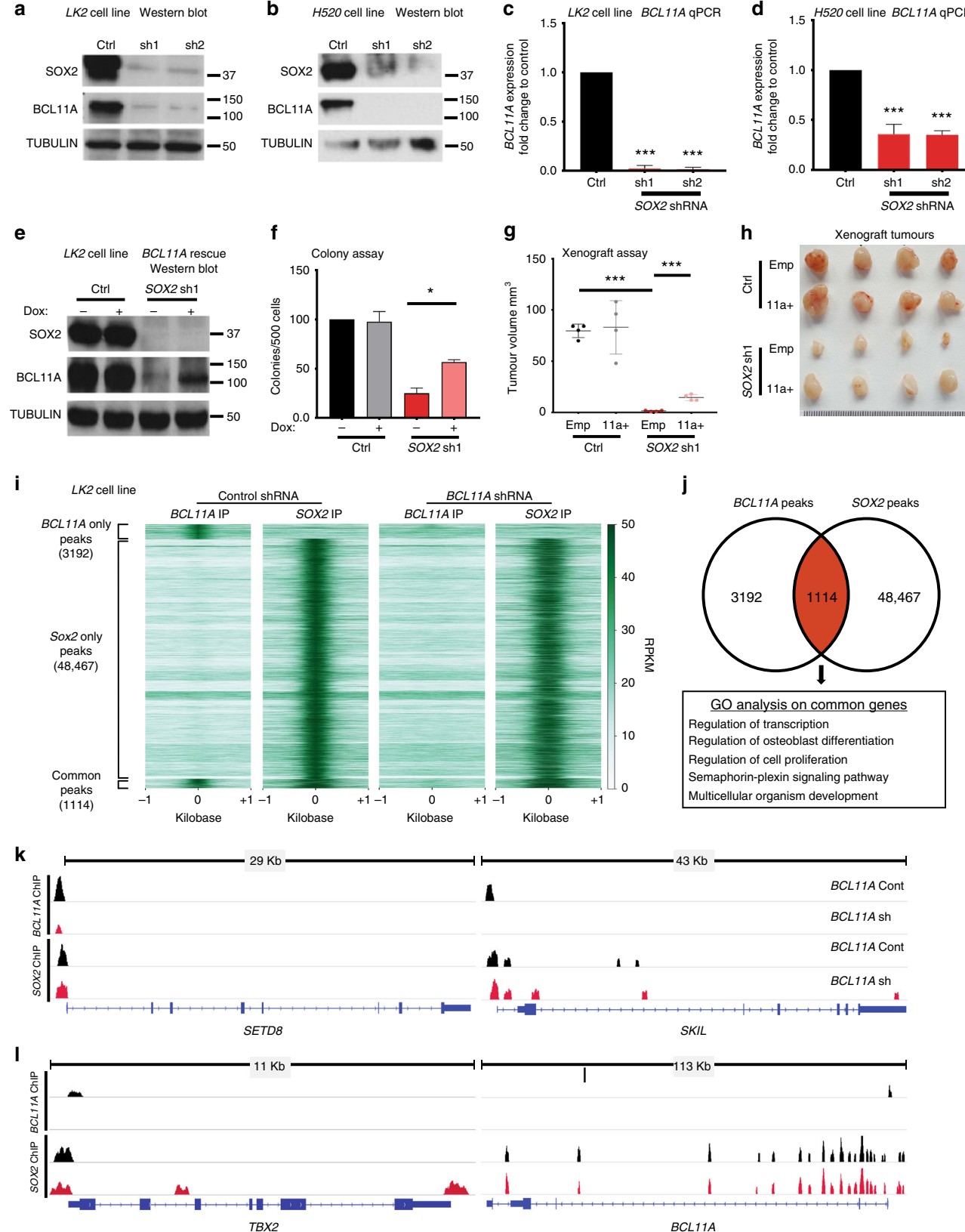

both basal stem cell and epithelial cell compartment from the trachea of *BCL11A*$^{ovx}$ mice (Fig. 4g–k and l–p). These results provide the first evidence that BCL11A regulates *Setd8* and *Skil* independent of SOX2-driven mechanisms.

We then investigated the expression of *SETD8*, *SKIL* and *TBX2* in *BCL11A*-KD or *SOX2*-KD cells and found a modest reduction in the expression of these genes in *BCL11A*-KD, which was more pronounced in *SOX2*-KD cells (Supplementary Fig. 10c–n). We

**Fig. 3** BCL11A and SOX2 occupy independent and common loci in the genome of LUSC cells. **a** and **b** Western blot showing SOX2 and BCL11A expression in *SOX2-KD* in LK2 (**a**) and H520 (**b**) cells transfected with control (scramble), shRNA1 or shRNA2 vectors. **c** and **d** *BCL11A* expression in *SOX2-KD LK2* (**c**) and *H520* (**d**) cells. **e** Western blot showing BCL11A rescue in *SOX2-KD* cells. Doxycycline (Dox) inducible *BCL11A* overexpression vector was transfected into control and *SOX2* shRNA1 LK2 cells and Dox treatment was performed for 48 h. **f** Graph depicting 3D matrigel assay in control, *SOX2-KD* and *BCL11A* rescue cells indicating a partial rescue in *SOX2-KD*, *BCL11A* overexpressing cells. **g** Graph indicating partial rescue of tumour size from *BCL11A* overexpressing *SOX2-KD* cells injected subcutaneously. The whiskers indicate the range of the data and the line represents the median. **h** Images of actual tumours measured. Four mice per cell line were monitored for 15 days after which tumours were removed and measured. Data presented as mean ± s.d. (n = 4). One-way ANOVA test performed, *p < 0.05, **p < 0.005, and ***p < 0.001. **i** LK2 cell line either transfected with control or shRNA1 vectors were used for BCL11A and SOX2 ChIP-Seq. Heatmaps showing BCL11A only, SOX2 only or common peaks in BCL11A or SOX2 IP in control and *BCL11A*-KD cells. **j** Venn diagram indicating the overlap of BCL11A and SOX2 target genes in LK2 cells. BCL11A target genes were derived by BCL11A IP in LK2 control cells. SOX2 target genes were derived from SOX2 IP in LK2 control cells. Image below show top five biological GO terms from GO analysis performed using DAVID. **k** and **l** IGV genome browser views for *SETD8, SKIL, TBX2* and *BCL11A*

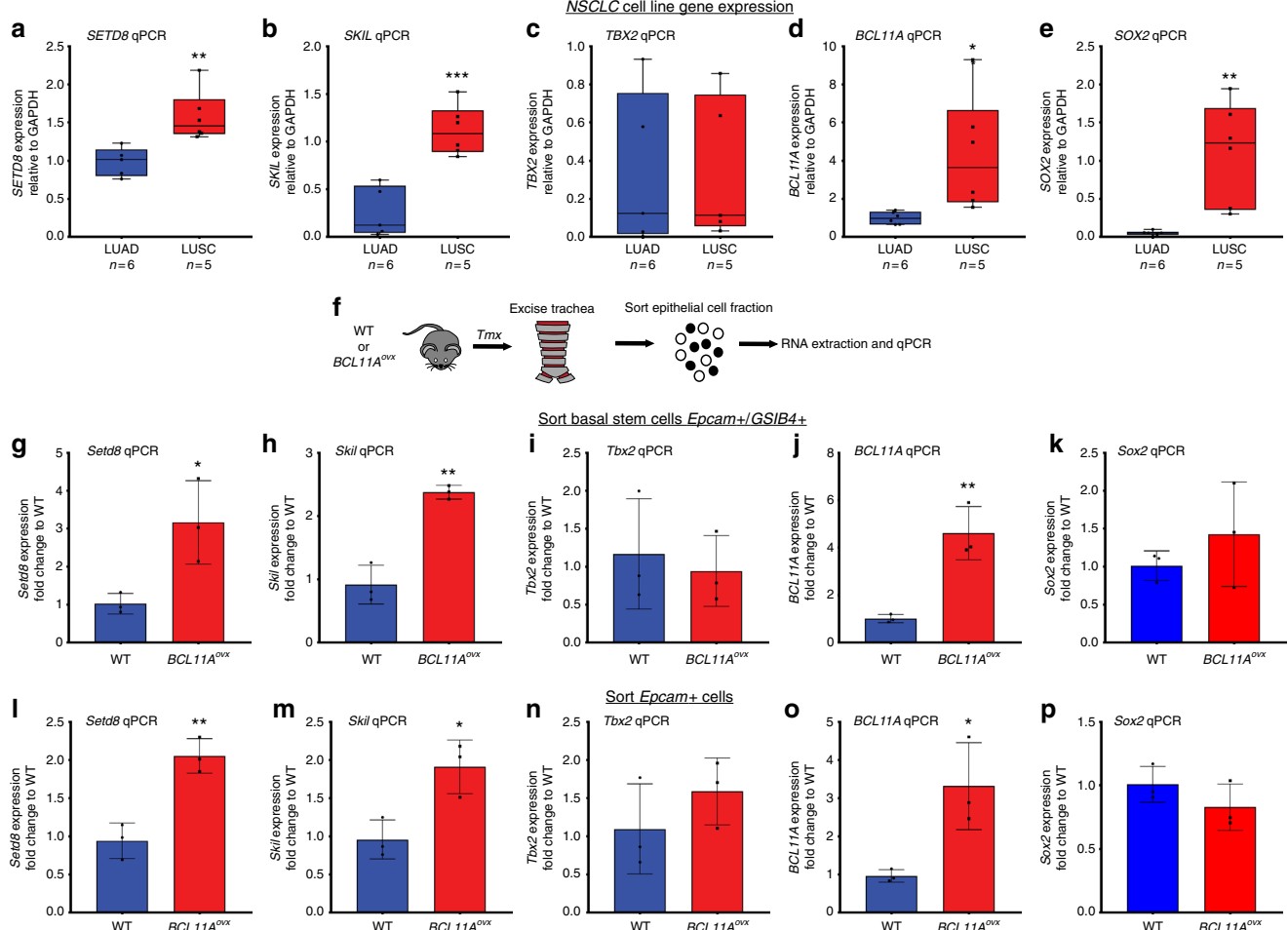

**Fig. 4** BCL11A regulates both *SETD8* and *SKIL* gene expression. SETD8 (**a**), SKIL (**b**), TBX2 (**c**), BCL11A (**d**), SOX2 (**e**) gene expression in NSCLC cell lines. **f** Sorting strategy for RNA extraction from WT or *BCL11A^OVX* mice treated with tamoxifen. SETD8 (**g**), SKIL (**h**), TBX2 (**i**), BCL11A (**j**), SOX2 (**k**) gene expression in basal stem cells. SETD8 (**l**), SKIL (**m**), TBX2 (**n**), BCL11A (**o**), SOX2 (**p**) gene expression in epithelial Epcam^+ve cells. The whiskers in all box plots represent minimum and maximum points. The box extends from 25th to 75th percentile and the line represents the median. Data presented as mean ± s.d. (LUSC n = 5 and LUAD n = 6; WT n = 3 and BCL11A^OVX n = 3). Student's *t*-test performed, *p < 0.05, **p < 0.005, and ***p < 0.001

reasoned that this could be due to SOX2 redundantly regulating *Setd8* in *BCL11A-KD* cells.

**SETD8 KD selectively inhibits LUSC tumour growth.** Collectively, our results thus far suggest that disrupting the BCL11A-SOX2 transcriptional programme could be selectively detrimental to LUSC cells. To test this hypothesis we employed a dox-inducible shRNA system[32], to KD *SETD8, SKIL* and *TBX2* in two LUSC cell lines and one LUAD cell line (H520

Supplementary Fig. 11a, b, and c; LK2 Supplementary Fig. 11d, e, and f; H1792 Supplementary Fig. 11g, h, and i). We found that SETD8-KD had a pronounced effect on colony formation selectively in LUSC cells but not in LUAD cells (Fig. 5a, e, i). SKIL-KD, also reduced colony formation but affecting both LUSC and LUAD cells suggesting its importance in NSCLC (Fig. 5b, f, g). In contrast, TBX2-KD had no effect on either LUSC or LUAD cells suggesting that it does not play a key role in the pathology of these cells (Fig. 5c, g, k).

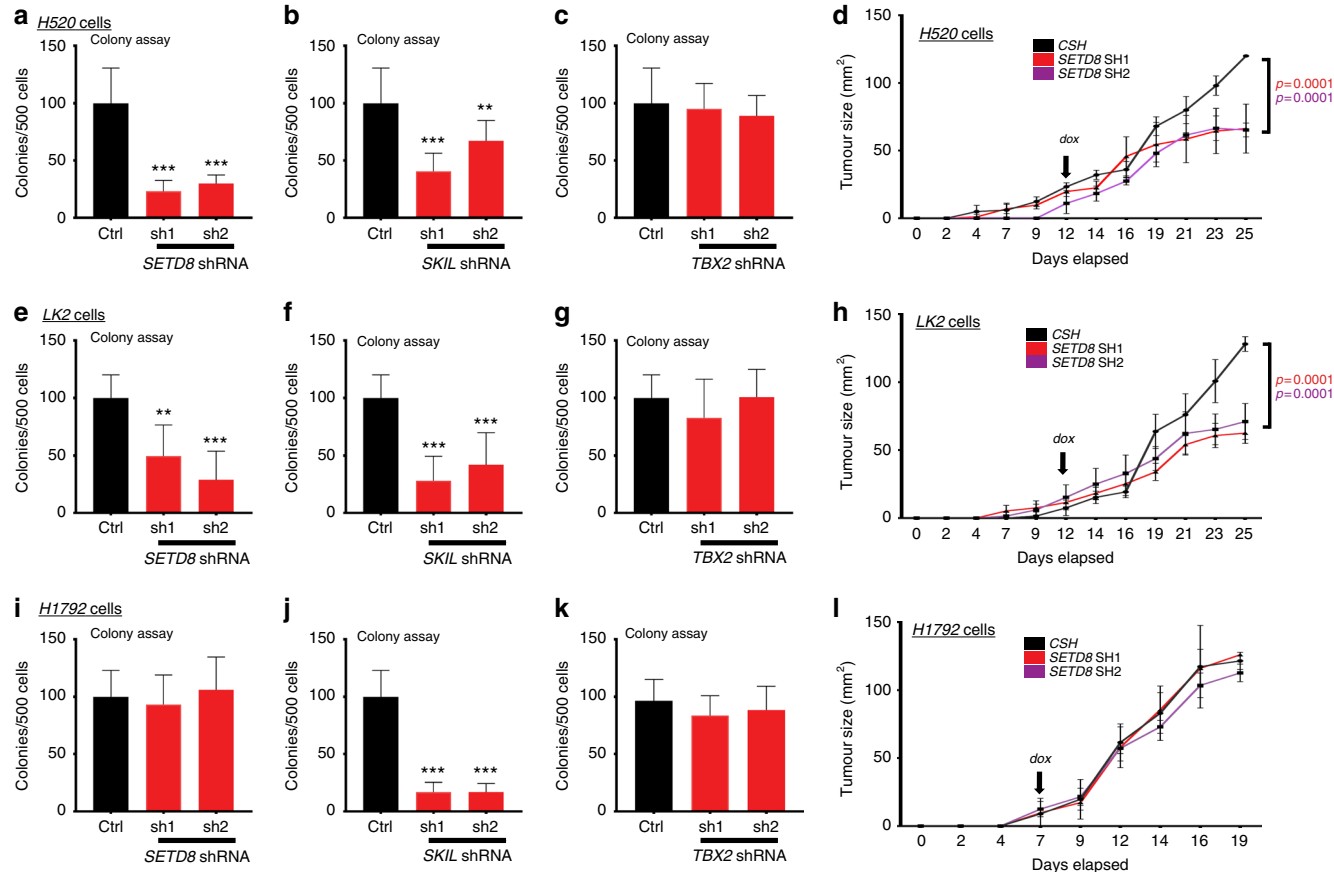

**Fig. 5** SETD8-KD has a selective effect on LUSC cell line oncogenic capacity. **a–c** Comparison of colony numbers in 3D matrigel assay from control, shRNA1 or shRNA2 against *SETD8* (**a**), *SKIL* (**b**) and *TBX2* (**c**) in H520 cells. **d** Tumour kinetics of control or SETD8-KD H520 cells injected into immune compromised mice. Arrow indicates administration of doxycycline. **e–g** Comparison of colony numbers in 3D matrigel assay from control, shRNA1 or shRNA2 against *SETD8* (**e**), *SKIL* (**f**) and *TBX2* (**g**) in LK2 cells. Data presented as mean ± s.d. ($n = 3$). **h** Tumour kinetics of control or SETD8-KD LK2 cells injected into immune compromised mice. Arrow indicates administration of doxycycline. **i–k** Comparison of colony numbers in 3D matrigel assay from control, shRNA1 or shRNA2 against *SETD8* (**i**), *SKIL* (**j**) and *TBX2* (**k**) in H1792 cells. Data presented as mean ± s.d. ($n = 3$). **l** Tumour kinetics of control or SETD8-KD H1792 cells injected into immune compromised mice. Arrow indicates administration of doxycycline. Data presented as mean ± s.d. One-way ANOVA with post Dunnett test performed, *$p < 0.05$, **$p < 0.005$ and ***$p < 0.001$

Given the selectivity of the SETD8-KD on LUSC vs. LUAD, we decided to focus on this gene for further studies. SETD8 is a member of the SET domain containing family and is known to catalyse the monomethylation of histone H4 Lys20[26,27] which is involved in recruiting signalling proteins or chromatin modifications[33]. In agreement with the qPCR data from sorted *BCL11A^{ovx}* cells (Fig. 4) we found that SETD8 protein levels were upregulated in airways and hyperplastic lesions of *BCL11A^{ovx}* mice (Supplementary Fig. 12a, b). Furthermore, analysis of the TCGA datasets revealed that *SETD8* expression correlates with *BCL11A* and *SOX2* expression in LUSC patients (Supplementary Fig. 12c and d).

To test the impact of SETD8 on xenograft tumour growth we injected cells with either Scram or *SETD8* shRNA vectors into immune compromised mice. Once tumours reached 0.25 cm$^2$ in size, we supplemented the mouse diet with doxycycline to induce the *SETD8* KD and measured tumour growth periodically. This setup would also mimic a therapeutic intervention in patients presenting with LUSC tumours. We found that initial tumour growth rates were comparable between Scram and shRNA cells across all three cell lines (Fig. 5d, h and l and Supplementary Fig. 13a–c) However, upon the addition of doxycycline tumours from tumours formed by the *SETD8-KD* cells slowed down significantly (Fig. 5d, h, and l). At the end of the experiment,

tumours formed by the *SETD8-KD* were ~50% smaller in size compared to Scram cells (Fig. 5d, h and l and Supplementary Fig. 13a–c). Interestingly, this effect was only found in the LUSC cell lines and not the LUAD which, is in agreement with the colony assays results.

**SETD8 inhibition sensitises LUSC cell lines to chemotherapy.** To expand our analysis further, we tested the effect of a SETD8 inhibitor, NSC663284[34], on 11 NSCLC cell lines (5 LUSC and 6 LUAD) in vitro. We treated all the NSCLC cell lines with a range of NSC663284 concentrations (full details of setup in Materials and methods) for 72 h and measured cell viability. Remarkably, we found that LUSC cells had significantly lower IC$_{50}$ (average 0.30 μM) compared to LUAD cells (average 7.65 μM) (Fig. 6a, b). To understand if SETD8 inhibition would add a clinical benefit to patients, we tested the effect of combining NSC663284 and cisplatin. First, we found that cisplatin treatment for 24 h alone affected LUSC and LUAD in a similar way with both cell types demonstrating a similar IC$_{50}$ (LUAD = 64.12 μM and LUSC = 31.67 μM) (Fig. 6c, d). However, if we pre-treat NSCLC cell lines with NSC663284 for 48 h and then combine cisplatin with NSC663284 for a further 24 h we found that LUSC (IC$_{50}$ = 4.66 μM) cells are more sensitive to cisplatin than LUAD cells

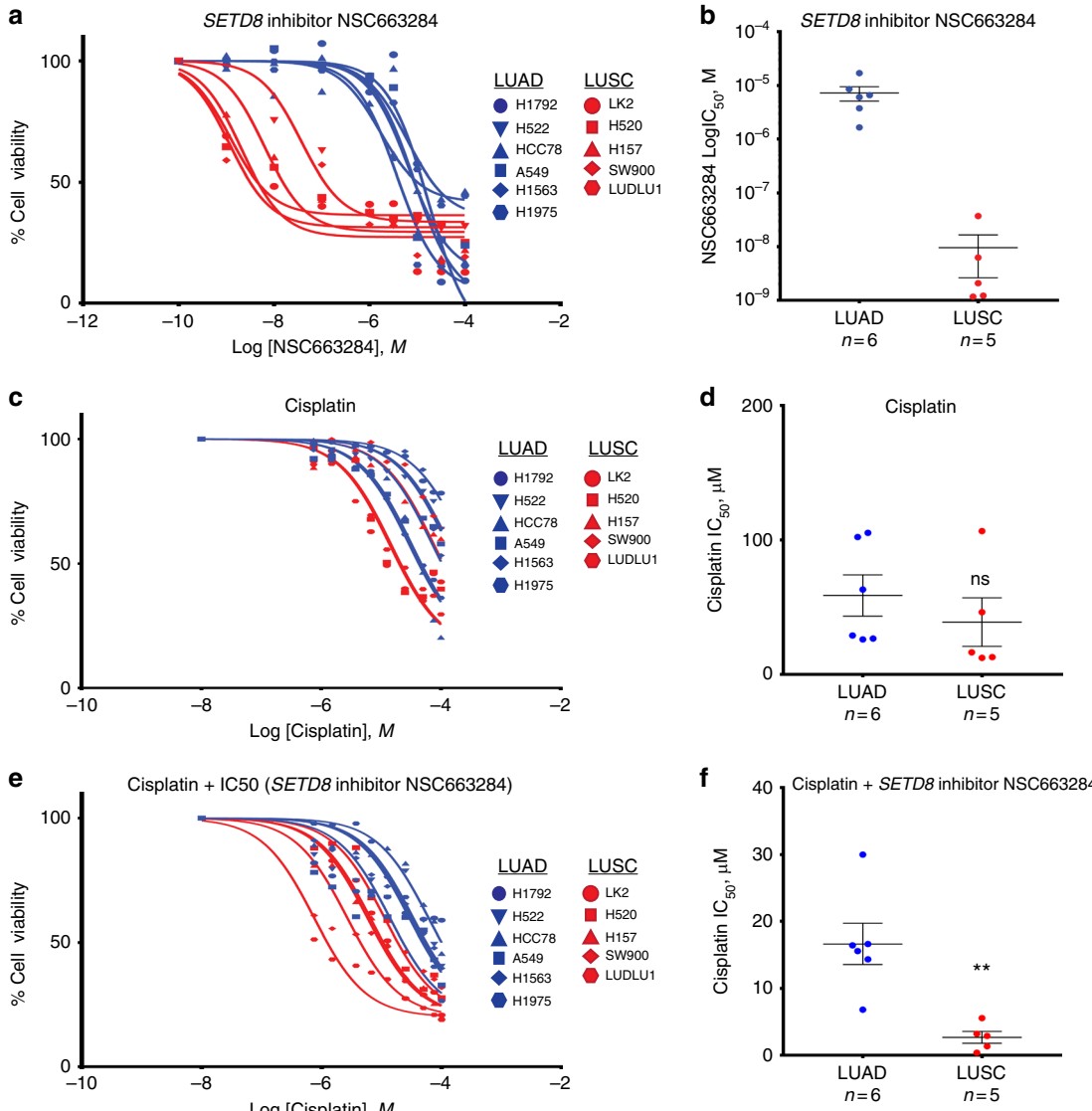

**Fig. 6** SETD8 inhibition preferentially sensitises LUSC cell lines to chemotherapy. **a** Dose–response curves were derived by treating NSCLC cell lines with SETD8 inhibitor NSC663284. 1000 cells were seeded and allowed to recover for 24 h. The inhibitor was then added at increasing concentrations to LUSC (red) and LUAD (blue) cells and Cell Titre (see Methods) assay was performed after 72 h. **b** $IC_{50}$ values were derived from the dose–response assay indicating LUSC cells are significantly more responsive to SETD8 inhibition than LUAD cells. **c** Dose–response curves were derived by treating NSCLC cell lines with cisplatin as above. **d** $IC_{50}$ values were derived from the dose–response assay indicating cisplatin effects LUSC and LUAD cells equally. **e** Dose–response curves derived from treating NSCLC cell lines with cisplatin and NSC663284 $IC_{50}$ concentration for each cell line as above. **f** $IC_{50}$ values were derived from the dose–response assay indicating SETD8 inhibition preferentially enhances cisplatin efficacy in LUSC cells. The whiskers indicate the range of the data and the line represents the median. Data presented as mean ± s.d. (LUSC $n = 5$ and LUAD $n = 6$). Student's $t$-test performed, $*p < 0.05$, $**p < 0.005$ and $***p < 0.001$

($IC_{50} = 32.21$ μM) (Fig. 6e, f). It is important to note that NSC663284 has also been reported as an inhibitor of Cdc25[35–37]. However, our shRNA data which is SETD8 specific corroborate the NSC663284 data and strongly supports the case for exploring SETD8 as a novel target for LUSC treatment alone or in combination with other chemotherapeutics.

## Discussion

Currently there is an unmet clinical need for patients with LUSC. Understanding and developing therapies for LUSC lags behind LUAD largely due to the association between tobacco smoking and LUSC incidences. Indeed, cancer genomics studies such as TCGA and TRACERx show that generally LUSC have a

mutational signature associated with smoking[12,17]. However, even with the decrease in smoking rates LUSC incidences are still high[38], reinforcing the calls for better molecular understanding and the development of new therapies to tackle the disease. In this study, we have demonstrated that the transcription regulator, *BCL11A* is upregulated in LUSC vs. LUAD and that it is a direct target of *SOX2*. BCL11A was initially discovered as an oncogene in B-Cell lymphomas[39] and we have recently reported that it is as an oncogene in TNBC[13]. We show that *BCL11A* upregulation leads to early stages of tumour development in the mouse and is critical for tumour maintenance even in the presence of SOX2. Our data suggests that disrupting the BCL11A-SOX2 transcriptional programme might provide a selective therapeutic window for LUSC patients. This was demonstrated by the sensitivity of

LUSC cell lines to the inhibition of *SETD8*, a gene we identified to be regulated by both BCL11A and SOX2 specifically in LUSC. SETD8 is a member of the SET domain containing family and is known to catalyse the monomethylation of histone H4 Lys20[26,27], which is involved in recruiting signalling proteins or chromatin modifications[33]. In addition, SETD8 has been shown to play a role in maintaining skin differentiation[40] and is dysregulated in multiple cancer types[41–43]. We found that *Setd8* was also upregulated in the mouse model of *BCL11A* overexpression in the lung. It will be important in the future to investigate the downstream targets of Setd8 and their role in LUSC development. In conclusion, our results describe an oncogenic role for *BCL11A* in LUSC and provide a potential future framework for tackling the unmet clinical need for LUSC patients.

## Methods

**Mouse models.** All mice used in this study were maintained at the Sanger Institute or the University of Cambridge. Housing and breeding of mice and experimental procedures were performed according to the UK 1986 Animals Scientific Procedure Act and local institute ethics committee regulations. The *BCL11A^ovx^* allele was generated following a strategy previously described[44]. Briefly, the *ROSA26* allele was targeted with a construct containing human *BCL11A* cDNA preceded by a loxP flanked STOP cassette and marked *eGFP* under the control of an internal ribosomal entry site (IRES) downstream of the inserted cDNA and transgene transcription is controlled by the *CAGG* promoter. The generation of the *Bcl11a^cko^* mice was described previously[16]. All mice were 8–12 weeks of age at the time of experiments, and at least three mice per cohort were used in each experiment. The primers used for genotyping are listed in Supplementary Data 4.

**Tracheal and BASC isolation and tracheosphere culture.** Tracheae were incubated in 50 U ml⁻¹ dispase (Sigma) for 45 min at 37 °C. 10 ml PBS was injected through each trachea using a 25 G 5/8″ needle to flush out sheets of epithelial cells. Cells were incubated in 0.25% trypsin for 5 min at 37 °C.

For isolating basal stem cells, mice were first injected intraperitoneally with tamoxifen (Sigma) dissolved in corn oil. Trachea was isolated as described above and cells were stained in PBS + 10%FBS with 1:500 anti-EpCAM BV421 (Biolegend) and GSIβ4-Biotin. BCs were considered Epcam^+ve^/GSIB4^+ve^ [30,31]. Cells were sorted directly into RLT lysis buffer for mRNA isolation.

For tracheosphere culture, cells were stained in PBS + 10%FBS + 1:100 anti-EpCAM PE-Cy7 (Biolegend) + 1:5000 DAPI on ice for 20 min. Live EpCAM^+ve^ cells were isolated using a MoFlo sorter. 2500 cells were plated in 100 µl 1:1 mix of mouse tracheal epithelial cell (MTEC)/plus media which is DMEM-Ham's F-12 (1:1 vol/vol), 15 mM HEPES, 3.6 mM sodium bicarbonate, 4 mM L-glutamine, 100 U ml⁻¹ penicillin, 100 µg ml⁻¹ streptomycin, and 0.25 µg ml⁻¹ fungizone; supplemented with 10 µg ml⁻¹ insulin, 5 µg ml⁻¹ transferrin, 0.1 µg ml⁻¹ cholera toxin, 25 ng ml⁻¹ epidermal growth factor (Becton-Dickinson, Bedford, MA, USA), 30 µg ml⁻¹ bovine pituitary extract, 5% FBS, and freshly added 0.01 µM retinoic acid[45] and growth factor reduced-Matrigel (Corning) per 24-well insert, in duplicate with 500 nM 4-hydroxy tamoxifen (Sigma) or ethanol (vehicle) for each mouse, and cultured for 15 days.

**Cell lines.** LK2, NCI-H520 and LUDLU1 cells were generously gifted to us by Dr. Frank McCaughan. NCI-H157, SW900, NCI-H1792, NCI-H522, NCI-HCC78, A549, NCI-H1563 and NCI-H1975 were obtained from the Sanger Cancer Project. All these cells were all maintained in RPMI 1640 (Gibco), 10% FCS and 1% Penicillin/streptomycin in a 37 °C incubator with 5% CO₂. All cell lines were routinely tested for mycoplasma.

**ShRNA-mediated KD.** *BCL11A* shRNA sequences were obtained from TRC consortium (TRCN0000033449 and TRCN0000033453) and cloned into a *piggyBac* transposon vector (PB-H1-shRNA-GFP) as describe previously[13]. Sox2 shRNA sequences (TRCN0000355694 and TRCN0000257314) were also cloned as above. H520, H1792 and LK2 cells were transfected with 4 µg of respective vector using Lipofectamine 3000 or Lipofectamine LTX (Invitrogen). Cells were treated with G418 (400 µg ml⁻¹) (Gibco) for 5 days after which GFP^+ve^ cell were sorted using the Sony SH800Z sorter and cultured.

**Doxycycline inducible shRNA-mediated KD.** SETD8 shRNA (TRCN0000359304 and TRCN0000359373), SKIL shRNA (TRCN0000431894 and TRCN0000424201); and TBX2 shRNA (TRCN0000232147 and TRCN0000232146) were cloned into pLKO-Tet-On vector[32]. Lentiviruses were generated by co-transfecting HEK293T cells with 3 µg of shRNA-encoding plasmid and 1 µg each of pMD2.G, pMDLg/pRRE and pRSVRev plasmids using Lipofectamine LTX. Growth media was exchanged after 5 h and lentivirus-containing supernatant was harvested after 48 h. The supernatant was filtered using a 0.45 µm filter cartridge and then Lenti-X

concentrator was added. The solution was incubated O/N at 4 °C and then centrifuged at 1500×*g* for 45 min. The pellet was then resuspended in 1/20th of the original volume using RPMI media. Cells were then infected with the virus for 48 h and selected in 1 µg ml⁻¹ puromycin. Cells infected with virus were grown in RPMI supplemented with 10% Tet-spproved FBS (Clontech) and 1 µg ml⁻¹ puromycin. shRNA expression was induced by culturing cells in the presence of 1 µg ml⁻¹ doxycycline for 72 h.

**Transfection and 3D colony assays.** The control or the *BCL11A* overexpression piggybac vectors were delivered into NSCLC cells using Lipofectamine LTX (Invitrogen). Transfected cells were maintained for 48 h and then cultured in puromycin (1 µg ml⁻¹). To induce BCL11A expression in LK2 *SOX2-KD* cells, doxycycline (Clonetech) was used at a final concentration of (1 µg ml⁻¹). 3D colony assays were performed by suspending 500 cells in matrigel (BD Biosciences) and seeding this cell-matrigel suspension onto a six-well plate. The plate was then incubated for 15 min in 37 °C/5% CO₂ to allow hardening of suspension. Growth media was added to the well and changed every 2–3 days for 20 days. All experiments were performed in triplicates.

**Preparation of RNA.** RNA was extracted using the RNeasy mini kit (Qiagen). Cell cultures in T25 flasks were first washed with cold PBS, and 350 µl of RLT was added. Cells were scraped, passed through a 20 G syringe five times and RNA was extracted using the RNeasy mini kit (Qiagen) according to manufacturer instructions. DNA was degraded by adding 20U Rnase-free DnaseI (Roche) for 30 min at room temperature. DnaseI treatment was performed on columns.

**Preparation of cDNA and qRT-PCR.** 1 µg of total RNA was diluted to a final volume of 11 µl. 2 µl of random primers (Promega) were added after which the mixture was incubated at 65 °C for 5 min. A master mix containing Transcriptor Reverse Transcriptase (Roche), Reverse Transcriptase buffer, 2 mM dNTP mix and RNasin Ribonuclease Inhibitors (Promega). This mixture was incubated at 25 °C for 10 min, then 42 °C for 40 min and finally 70 °C for 10 min. The resulting cDNA was then diluted 1:2.5 in H₂O for subsequent use. qPCR was performed using a Step-One Plus Real-Time PCR System (Thermofisher Scientific). Either Taqman (ThermoFisher Scientific) probes with GoTaq Real Time qPCR Master Mix (Promega) or primers (Sigma) with PowerUp SYBR Green Master Mix (ThermoFisher Scientific) were used. All probe and primer details can be found in Supplementary Data 5 and 6. The enrichment was normalised with control mRNA levels of GAPDH and relative mRNA levels were calculated using the ΔΔCt method comparing to control group.

**Western blot.** Cells were lysed using RIPA (Cell signalling) and protease inhibitors (Roche) as per manufacturer instructions. Total protein was measured using the bicinchoninic acid (BCA) method (Pierce Biotechnology). In total, 50 mg cell lysate was separated using 7.5% SDS–PAGE gels and transferred to PVDF membranes by electro-blotting. Membranes were blocked in 5% (w/v) milk in Tris-buffered saline containing 0.1% Tween-20 (TBST). Blots were then incubated at 4 °C overnight with primary antibodies as indicated, washed in TBST and subsequently probed with secondary antibodies for 1 h at room temperature. ECL solution was then added to the membrane and analysed. Antibodies used were, anti-BCL11A (ab191401, Abcam, 1:3000), anti-SOX2 (ab97959, Abcam, 1:2000) and anti-TUBULIN (ab7291, Abcam, 1:10000). All the original western blot images can be found in Supplementary Figures 14 and 15.

**Co-immunoprecipitation.** Cells were lysed using RIPA (Cell signalling) and protease inhibitors (Roche) as per the manufacturer's instructions. Total protein was measured using the BCA method as above (Pierce Biotechnology). Briefly, 500 µg cell lysates were pre-cleared for 3 h at 4 °C to remove nonspecific binding. Then, the pre-cleared lysates were incubated with anti-BCL11A (Bethyl, A300–382A) and SOX2 (R&D Systems, AF2018) or control IgG at 4 °C overnight. Next day 50 µl of Dynabeads Protein G (Thermo Fisher Scientific) were added to each sample. After 3 h, the complex was washed three times with RIPA buffer, and then analysed by Western Blot performed as described above.

**Histology and immunohistochemistry and immunofluorescence.** Cultured organoid were fixed with 4% paraformaldehyde in PBS for 4 h at room temperature. After rinsing with PBS, fixed colonies were immobilised with Histogel (Thermo Scientific) for paraffin embedding. 5 µm sections of lung tissues or embedded colonies were stained with haematoxylin and eosin (H&E) and immunostained with antibodies for BCL11A (IHC—ab191402, Abcam, 1:1000), SOX2 (IHC—ab97959, Abcam, 1:1000; IF—14-9811-82, eBioscience, 1:200), Ki67 (IHC—MA5-14520, Thermo Scientific, 1:1000; IF—ab16667, Abcam, 1:300), GFP (IF—ab13970 Abcam, 1:1000), Keratin 8 (IF—TROMA-I, DSHB, 1:100), Keratin 5 (IHC—ab52635, Abcam, 1:1000; IF—905501, Biolegend, 1:1000), P63 (IHC—ab735, Abcam, 1:200; IF—ab735, Abcam, 1:200), CCP10 (IHC—sc25555, Santa Cruz Biotechnology, 1:500), SETD8 (IF—ab3798, Abcam, 1:100). IHC secondary staining involved an HRP-conjugated donkey anti-rabbit or donkey anti-mouse secondary (1:250, Thermo Scientific) and were detected using DAB reagent

(Thermo Scientific). IF secondary staining involved goat anti-chicken 488, goat anti-rabbit 647, goat anti-rat 647, goat anti-rabbit 488 and goat anti-mouse 647 (1:2000, Invitrogen). Nuclear stain was detected using Haematoxylin (IHC) or ProLong Gold Antifade Mountant DAPI (Thermofisher, P36941) (IF). IHC images were acquired using a Zeiss Axioplan 2 microscope and IF images were acquired using a Leica TCS SP5 confocal microscope and analysed on Image J.

**Xenograft tumour assays**. One million H520 and LK2 cells were suspended in 25% matrigel (BD Biosciences) and HBSS. This mixture was subcutaneously injected in 6–12-week-old NSG mice. To induce BCL11A overexpression in *SOX2-KD* xenografts or to induce shRNA expression, mice were fed doxycycline pellets (Envigo, TD.01306, 625 mg/kg). Mice were randomised to receive injections of either control or shRNA-KD cells in the xenograft experiment. Tumours were measured blindly by animal technicians who did not know what was injected into the specific mouse. Mice were culled as specified in figure legends and resulting tumours were analysed.

**ChIP-Seq and ChIP-qPCR**. ChIP-Seq experiments were performed as described[46]. Antibodies used were BCL11A (Bethyl, A300-382A) and SOX2 (R&D Systems, AF2018). Briefly 2 × 15 cm plates per cell line were formaldehyde crosslinked, nuclear fraction was isolated and chromatin sonicated using Bioruptor Pico (Diagenode). IP was performed using 100 μl of Dynabeads Protein G (Thermo Fisher Scientific) and 10 μg of antibody. The samples were then reverse crosslinked and DNA was eluted using Qiagen MinElute column. Sample was then processed either for sequencing or qPCR. Primers used for qPCR are listed in Supplementary Data 6.

**Drug assays**. SETD8 inhibitor NSC663284 (Cayman Chemical Company, 13303) was suspended using DMSO in 10 mM stock concentration. Cisplatin (LKT Laboratories, C3374) was suspended in 154 mM NaCl at a 3 mM stock concentration. Cells were cultured as above and seeded at 1000 cells per 96-well plate and left to recover for 24 h. The edges of the 96-well plate were avoided to ensure accuracy in measurement. For NSC663284, an initial dilution of 1:100 from stock was performed in RPMI media for the first concentration of 10−4 M. Half log dilutions were performed in RPMI media reaching 10−6 M and after which full log dilutions were performed reaching 10−10 M. For cisplatin, initial dilution in RPMI media were made to achieve 100 and 75 μM. The 100 μM solution was used to make the following solutions 50, 25, 12.5, 6.75, 3.75, 1.5, and 0.5 μM. Cells were treated with vehicle for 48 h after which the above doses of cisplatin were added. For cisplatin + NSC663284 experiment the NSC663284 $IC_{50}$ concentration for each cell line was calculated and added initially for 48 h and then added again along with the cisplatin dilution series as above for 24 h. Data analysis for drug inhibitor assays was performed in GraphPad Prism 7.02 (San Diego, CA). Data were fitted to obtain concentration–-response curves using the three-parameter logistic equation (for pIC50 values). Emax was constrained to 100% while the basal (Emin) parameter was contrained to 20%. Statistical differences were analysed using one-way ANOVA or Student's *t*-test as appropriate with post hoc Dunnett's multiple comparisons, and $p < 0.05$ was considered significant.

**BCL11A IHC on patient tumours**. TMAs contained LUAD ($n = 99$) and LUSC ($n = 120$) cases of archival primary pulmonary tumours collected under East Midlands NRES REC approved project (ref. 14/EM/1159). 1 mm ($n = 3$) cores are present per case which were initially scored (average nuclear staining intensity) as 0 = neg, 1 = weak, 2 = moderate, 3 = strong. A median score was calculated for each case and re classified as 0 = neg, 1 = moderate, 2 = strong. All tissues and data are anonymised to the research team. IHC was performed using BCL11A antibody (ab19487, 1:200) with CC1 antigen retrieval using a Ventana discovery xt. Digital images of stained TMAs were scanned with a Nanozoomer RX instrument and scored on-screen by MD.

**TCGA gene expression analysis**. The TCGA data was accessed from the recount2 database[47] containing gene level count data from RNA-seq and clinical data from primary tumour samples from patients diagnosed with LUAD and LUSC, respectively. EdgeR[48] was used to test for differential expression of transcription factors (as defined by Tfcheckpoint.org). For this, compositional differences between samples were normalised using the trimmed median of M-values method and a gene-specific dispersion was estimated for each gene. A negative binomial generalised log-linear model was fit to each gene with the covariates "plate + disease_type" (for the LUAD versus LUSC comparison) or "patient + disease_type" (for the tumour versus normal comparison). A log-likelihood ratio test was conducted to test whether the coefficient of the disease_type variable is non-zero, followed by Benjamini–Hochberg adjustment of *p* values to account for multiple testing. Plate A277 and A278 from the LUAD dataset were removed prior to analysis as they showed a clear separation along the first component in a principal component analysis from all other LUAD samples.

**ChIP-Seq analysis**. ChIP-Seq libraries were sequenced on illumina Hiseq2000 platform at the Wellcome Sanger Institute. Each library was divided into two and

sequenced on different lanes. Reads were subsequently run through a pipeline at the sequencing facility to remove adaptor sequences and align to the reference genome among others. Alignment was done using mem algorithm in BWA (version 0.7.15) and human_g1k_v37 was used as the reference genome. Aligned and processed reads were received as compressed CRAM files. Samtools (version 1.3.1) was used to decompress the CRAM files and filter uniquely mapped reads in proper pairs. Next, reads from the two runs were combined into a single BAM file using 'merge' function in samtools. Bedtools intersect was then used to remove reads falling into blacklisted genomic regions or unplaced genomic contigs of the GRCh37 assembly before marking and removing duplicate reads using MarkDuplicates function in Picard tools. Next, DownsampleSam in picard tools was used to sample ~105 million reads from each BAM file. Significantly enriched genomic regions relative to input DNA were identified using MACS2 (version 2.1.1.20160309) with *p*-value cutoff of 1.00e−05. Heatmaps generation: Mapped read counts were calculated in a 10 bp window and normalised as reads per kilobase per million (RPKM mapped reads) using bamCoverage module from deeptools (version 2.5.1)[49]. This coverage file was used to compute score matrix ± 1 kb around peak summits using computeMatrix reference-point module (from deeptools version 2.5.1)[49]. Heatmaps of binding profiles around peak summits were then generated using plotHeatmap module in deeptools (version 2.5.1)[49]. Number of overlapping peaks between BCL11A and SOX2 and nearest downstream genes to peaks were determined using ODS and NDG utilities, respectively, in PeakAnnotator (version 1.4). For annotating nearest downstream genes, Homo sapiens GRCh37 (release 64) from ensembl was used.

**Code availability**. Source code will be available on request.

**Data availability**. The Chipseq data is available from ArrayExpress (accession numbers: # E-MTAB-6958) . The authors declare that all remaining data supporting the findings of this study are available within the article and its Supplementary Information files or from the authors upon request.

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

## Acknowledgements

We would like to thank the staff at Sanger Institute, Research Service Facility (RSF) for their assistance. We would like to thank Dr. Catherine Wilson and Dr. Deborah Burkhart (Department of Biochemistry, Cambridge) for her help with the Adenovirus experiment. We would like to thank Dr. Emma Rawlins for helpful discussions and comments. We would like to thank Agnetta Lazarus and members of the WTK Lab for comments on the manuscript. G.L. was supported by the BBSRC (Grant BB/M00015X/2). F.H. is funded by a Gates foundation Studentship. E.Z is funded by an AstraZeneca Studentship, K.B. is funded by a CCC studentship R.U. is funded by a NC3Rs studentship. F.M. and L.L.C. were funded by a Wellcome Trust Intermediate Clinical Fellowship to F.M.M (WT097143MA).W.T.K. is funded by a CRUK Career Establishment Award (C47525/A17348), CRUK Small Molecule Drug Discovery Project Award (C47525/A25850), The Isaac Newton Trust Grant (16.38c), University of Cambridge and Magdalene College, Cambridge.

## Author contribution

K.A.L. designed and performed the majority of the experiments and analysed most of the data. F.H. analysed the ChIP-seq data. E.Z. performed the ChIP-qPCR experiments. K.B. analysed the TCGA data. S.P. performed some cell line work. R.U. assisted with the BCL11A rescue experiment. M.F.S. performed Co-IP experiment. L.B. measured the airway thickness. L.S.C. characterised the lung pathology of adenovirus mice. G.L. designed and analysed the drug assays. J.K.W. and J.H.L. performed the tracheoshphere organoid experiment. L.L.C. and F.M. performed the mouse tissue IHC. M.D. and J.L.Q. performed and analysed the BCL11A IHC on patient tumours. P.L. assisted with the NGS sequencing and provided the *Bcl11a^cko* mice. Adenovirus Cre administration was performed under G.E. supervision. D.C. generated the *BCL11A^ovx* mice. W.T.K. conceptualised and supervised the study. K.A.L. and W.T.K. wrote the manuscript.

## Additional information

**Competing interests:** The authors declare no competing interests.

