## [Peer Review File · Nature Communications]

Reviewers' comments:

Reviewer #1 (Remarks to the Author):

In this manuscript, the authors have identified BCL11A as an important factor which together with SOX2 is can drive lung squamous cell carcinoma (LUSC) development and catalyze efficient growth of lung basal cell organoids. SOX2 appears to command BCL11A expression. Both factors bind to many sites on the DNA, with substantial overlap (not surprising in view of the many binding sites of SOX2 and BCL11A). The authors single out a number of genes (SETD8, SKIL and TBX2) that upon shRNA inhibition of BCL11A or SOX2 are downregulated. Given the inhibitory effect on SETD8 they further explored whether SETD8 inhibition can effectively inhibit LUSC growth and enhance killing by standard CsPt treatment. This appeared to be the case. The authors conclude that inhibition of SETD8 is worth further exploring therapeutically.

Critique

The study clearly shows the relevance of BCL11A and SOX2 in LUSC carcinoma development and the experiments shown are well-performed. However, they do not fully support the claims of this MS. How BCL11A -which is a primary focus of this study- acts and to what extent the synergism claimed between SOX2 and BCL11A in inhibiting downstream effectors is of relevance remains unclear. In addition, there are many loose ends, making the MS in its present form not suitable for publication in Nature Communications.

- The arguments for selecting of SETD8, TBX2, and SKIL as critical targets of Sox2/BCL11A need further substantiation. These genes have all been reported to have oncogenic activity in other studies and therefore an easy choice.
- The authors point to TBX2 and SKIL as other factors that might be involved. Have they assessed whether the inhibition of TBX2 and/or SKILL has a similar growth inhibitory effect?
- The inhibition of SETD8 by BCL11A shRNA is statistically significant but rather subtle in its actual effect. The very modest impairment of SET8 (as well as TBX2 and SKIL) expression could easily be the consequence of indirect effects of BCL11A downregulation. In contrast, depletion of SOX2 virtually abolishes SETD8 expression. Therefore, the claim that BCL11A is important for the expression of these genes is not supported by the presented data. It remains uncertain whether the oncogenic activity of BCL11A is mediated through these factors.
- Although SOX2 and BCL11A can be co-precipitated evidence is lacking that they form a complex that is critical for the cooperative regulation of expression of genes. At least for the targets discussed in this manuscript this is not convincingly shown.
- By using an SETD8 inhibitor (NSC663284) the authors show that this inhibitor is much more effective impairing proliferation of LUSC than LUAD cells, this in line with the more pronounced expression of SETD8 in LUSC. However, given the lack of specificity of NSC663284 and its reported inhibitory activity towards CDC25A it remains unclear whether it is the inhibition of SETD8 that make LUSC sensitive to NSC663284. The authors need to show that bringing levels of SETD8 down e.g. by a DOX regulatable shRNAs has a similar effect.
- They also need to show that what they observe in cell lines has relevance in vivo in xenograft experiments. Given the limited specificity of NSC663284 this is again probably best shown by inducible shRNAs against SETD8.
- Using the specific drug regimen in the cell lines tested –first treatment with NSC663284 followed by CsPt- requires a further explanation. Why was this regimen chosen? Does co-administration not show a similar effect? What brought the authors to this regimen?

- SETD8 has recently been reported to prevent p53 activation in neuroblastoma cells. Could a similar mechanism of action explain the increased sensitivity for CsPt after NSC663284 administration? Please mention the p53 status of the cell lines used in these experiments and indicate whether this could be of relevance.

Reviewer #2 (Remarks to the Author):

Lazarus et al. report that BCL11A is an oncogene for lung squamous cell carcinoma (LUSC). The authors further showed that BCL11A interacts with SOX2 to co-regulate genes including SETD8 in LUSC cell lines. Moreover, the authors showed that inhibition of SETD8 provides therapeutic gains in LUSC cell lines. The findings are interesting in that they provide a targetable BCL11A/SETD8 axis that is specifically activated in LUSC but not in lung adenocarcinoma cells (LUAD). The authors provide abundant molecular and cellular data to support this. That said, there are some concerns that need to be addressed.

1. What is BCL11A? What has been known for this protein in gene regulation?
2. The hyperplastic cell nodules developed in BCL11A overexpressing airways are interesting. High magnification picture of these nodules is required (Figure 2b). Do these cells attach to the basement membrane? E-cadherin staining will help. Do these nodules express K4, K13, genes associated with squamous cell metaplasia?
3. BCL11A staining pattern in Fig2e and supplemental Fig5 is not similar to the staining in Fig1d. A more specific staining is required for both Figures.
4. Overexpression of BCL11A in the airways with Ad-cre unlikely targets basal cells because there are extremely few basal cells in the small airways. Nevertheless, the authors suggest that LUSC originates from basal cells. The authors need to provide explanation. Alternatively, the authors can do targeted conditional overexpression of BCL11A in basal cells with KRT5-CreER alleles.
5. What is the expression pattern of SETD8 in the mouse model overexpressing BCL11A?

Minor: High SOX2 levels do not necessarily mean squamous differentiation. Supplementary Figure 3b). The normal proximal airways (e.g. trachea) have high levels of SOX2.

Dear reviewers,

Thank you for your comments and suggestions which have further strengthened our conclusions. Please find below our point-by-point responses to your comments.

Reviewer #1 (Remarks to the Author):

In this manuscript, the authors have identified BCL11A as an important factor which together with SOX2 is can drive lung squamous cell carcinoma (LUSC) development and catalyze efficient growth of lung basal cell organoids. SOX2 appears to command BCL11A expression. Both factors bind to many sites on the DNA, with substantial overlap (not surprising in view of the many binding sites of SOX2 and BCL11A). The authors single out a number of genes (SETD8, SKIL and TBX2) that upon shRNA inhibition of BCL11A or SOX2 are downregulated. Given the inhibitory effect on SETD8 they further explored whether SETD8 inhibition can effectively inhibit LUSC growth and enhance killing by standard CsPt treatment. This appeared to be the case. The authors conclude that inhibition of SETD8 is worth further exploring therapeutically.

Critique

The study clearly shows the relevance of BCL11A and SOX2 in LUSC carcinoma development and the experiments shown are well-performed. However, they do not fully support the claims of this MS. How BCL11A -which is a primary focus of this study- acts and to what extent the synergism claimed between SOX2 and BCL11A in inhibiting downstream effectors is of relevance remains unclear. In addition, there are many loose ends, making the MS in its present form not suitable for publication in Nature Communications.

- The arguments for selecting of SETD8, TBX2, and SKIL as critical targets of Sox2/BCL11A need further substantiation. These genes have all been reported to have oncogenic activity in other studies and therefore an easy choice.
- The authors point to TBX2 and SKIL as other factors that might be involved. Have they assessed whether the inhibition of TBX2 and/or SKILL has a similar growth inhibitory effect?

We based our choice of these three targets on two essential criteria. First, the proximity to genes encoding for transcription regulators, which narrowed down the list to 356 peaks and second, a reported dysregulation in NSCLC. SKIL and TBX2 have been reported to be upregulated in non-small cell lung cancer^{1,2}. SETD8 on the other hand has been indirectly linked to NSCLC as a key target of miR-382 in NSCLC³.

With regards to the biological effects of these three genes, we now include data in the revised manuscript to address this question. We have generated DOX-inducible shRNA cell lines (2 on target shRNAs per gene plus 1 Scram shRNA) for all three genes in (in 2 LUSC and 1 LUAD cell lines – that is 27 cell lines in total) (Figure 5 and Supplementary Figure 11). In summary, we found that:

- 1) TBX2 knockdown has no impact on any of the cell lines suggesting that it is not integral to NSCLC pathology;*
- 2) SKIL knockdown was detrimental to colony formation in both LUSC and LUAD cell lines suggesting that it is a common oncogenic factor in NSCLCs, and;*
- 3) SETD8 knockdown was detrimental to colony formation and xenograft growth in LUSC cell lines but not LUAD cells.*

These, results thus further support the justification of focusing on SETD8 in our study as a novel target of BCL11A and SOX2 in LUSC.

- The inhibition of SETD8 by BCL11A shRNA is statistically significant but rather subtle in its actual effect. The very modest impairment of SET8 (as well as TBX2 and SKIL) expression could easily be the consequence of indirect effects of BCL11A downregulation. In contrast, depletion of SOX2 virtually abolishes SETD8 expression. Therefore, the claim that BCL11A is important for the expression of these genes is not supported by the presented data. It remains uncertain whether the oncogenic activity of BCL11A is mediated through these factors.

We have shown in the manuscript that BCL11A expression is modulated by SOX2 as SOX2 knockdown downregulated BCL11A expression in LUSC cell lines. The ChIPseq data together with the knockdown data for both BCL11A and SOX2 support the notion that they regulate the expression of Setd8, Skil and Tbx2. To determine the

contribution of BCL11A to the transcriptional regulation of these three genes, we induced BCL11A overexpression in the absence of SOX2 upregulation using the BCL11A^{ovx} mice and isolated Tracheal Basal Stem Cells (TBSC) and Tracheal Epcam^{+ve} cells (from three independent mice) for qRT-PCR analysis. We found that BCL11A overexpression induced the expression of Setd8 and Skil but not Tbx2 (Figure 4). Therefore, this data indicates that BCL11A plays a direct role in regulating SETD8 and SKIL gene expression. In addition, we also show in the revised manuscript that knockdown of Setd8 and Skil are detrimental to LUSC cell lines (see colony assay above and xenograft data below). Collectively these results support the hypothesis that Setd8 and Skil are partially responsible for BCL11A and SOX2 oncogenic effects.

- Although SOX2 and BCL11A can be co-precipitated evidence is lacking that they form a complex that is critical for the cooperative regulation of expression of genes. At least for the targets discussed in this manuscript this is not convincingly shown.

We thank the reviewer for this comment and therefore, we have edited the text throughout to reflect this point. We agree that further detailed biochemical and biophysical experiments are needed to understand the nature of the protein-protein interaction between BCL11A-SOX2. We feel however, that this would warrant an independent study beyond the scope of this manuscript where we report the observation that BCL11A and SOX2 bind and regulate shared genomic loci in LUSC.

- By using an SETD8 inhibitor (NSC663284) the authors show that this inhibitor is much more effective impairing proliferation of LUSC than LUAD cells, this in line with the more pronounced expression of SETD8 in LUSC. However, given the lack of specificity of NSC663284 and its reported inhibitory activity towards CDC25A it remains unclear whether it is the inhibition of SETD8 that make LUSC sensitive to NSC663284. The authors need to show that bringing levels of SETD8 down e.g. by a DOX regulatable shRNAs has a similar effect.

- They also need to show that what they observe in cell lines has relevance in vivo in xenograft experiments. Given the limited specificity of NSC663284 this is again probably best shown by inducible shRNAs against SETD8.

We thank the reviewer for the comment and suggestion. As highlighted above we followed the reviewer's suggestion and generated DOX-inducible shRNA cell lines (2 shRNAs) for Setd8 in (in 2 LUSC and 1 LUAD cell lines). We report in the revised manuscript that SETD8 knockdown was only detrimental to the LUSC cell lines but not the LUAD cell line (Figure 5 in the revised manuscript). Therefore, our shRNA data which is SETD8 specific corroborate the NSC663284 data and strongly supports the case for exploring SETD8 as a novel target for LUSC treatment alone or in combination with other chemotherapeutics.

- Using the specific drug regimen in the cell lines tested –first treatment with NSC663284 followed by CsPt- requires a further explanation. Why was this regimen chosen? Does co-administration not show a similar effect? What brought the authors to this regimen?

Our initial observations with NSC663284 showed that treating for 72hours was optimal for effective response. We did not see any effect of the NSC663284 at either 24 or 48 hours, and this is most likely due to our hypothesis that inhibiting an epigenetic mechanism requires time. Therefore, we treated the cells with NSC663284 for 48hours and then Co-administered NSC663284 and Cisplatin for a further 24hours to capture the impact of SETD8 inhibition on this potent cytotoxic drug.

- SETD8 has recently been reported to prevent p53 activation in neuroblastoma cells. Could a similar mechanism of action explain the increased sensitivity for CsPt after NSC663284 administration? Please mention the p53 status of the cell lines used in these experiments and indicate whether this could be of relevance.

The cell lines used in this study are p53 mutant (COSMIC database, https://cancer.sanger.ac.uk/cell_lines) suggesting that the effects of SETD8 inhibition are p53 independent.

Reviewer #2 (Remarks to the Author):

Lazarus et al. report that BCL11A is an oncogene for lung squamous cell carcinoma (LUSC). The authors further showed that BCL11A interacts with SOX2 to co-regulate genes including SETD8 in LUSC cell lines. Moreover, the authors showed that inhibition of SETD8 provides therapeutic gains in LUSC cell lines. The findings are interesting in that they provide a targetable BCL11A/SETD8 axis that is specifically activated in LUSC but not in lung adenocarcinoma cells (LUAD). The authors provide abundant molecular and cellular data to support this. That said, there are some concerns that need to be addressed.

1. What is BCL11A? What has been known for this protein in gene regulation?

We thank the reviewer for this question and the comments for the manuscript. BCL11A is a zinc finger transcription factor first discovered as a translocated locus in a lethal paediatric B-cell chronic lymphocytic leukaemia⁴. Subsequently it was identified as a protooncogene in numerous B-cell malignancies. Recent studies have demonstrated an essential requirement for Bcl11a in B-cell lymphogenesis and as a transcriptional repressor of fetal haemoglobin⁵. In addition, BCL11A has been shown to be a component of various chromatin remodelling complexes^{6,7}

2. The hyperplastic cell nodules developed in BCL11A overexpressing airways are interesting. High magnification picture of these nodules is required (Figure 2b). Do these cell attach to the basement membrane? E-cadherin staining will help. Do these nodule express K4, k13, genes associated with squamous cell metaplasia?

We thank the reviewer for the comment. We now include high magnification pictures of the nodules (Figure 2b and see below). We also performed E-Cadherin staining and found no expression in the hyperplastic nodules (See below). With regard to markers of squamous cell metaplasia, we have included immunostaining for K5 and p63, both of which are known to be associated with the LUSC phenotype (Supplementary Figure 3b).

High magnification H&E image of nodule in BCL11Aovx mouse

E-cadherin immuostaining in control and BCL11Aovx mice.

3. BCL11A staining pattern in Fig2e and supplemental Fig5 is not similar to the staining in Fig1d. A more specific staining is required for both Figures.

We thank the reviewer for the comment. The difference in pattern is due to the different IHC protocols used to identify BCL11A in human tumour material vs. mouse organoid and lung tissue. Unfortunately, the tumour tissue was stained with ab19487, which is, a mouse monoclonal thus making it difficult to use on mouse tissue.

4. Overexpression of BCL11A in the airways with Ad-cre unlikely targets basal cells because there are extremely few basal cells in the small airways. Nevertheless, the authors suggest that LUSC originates from basal cells. The

authors need to provide explanation. Alternatively, the authors can do targeted conditional overexpression of BCL11A in basal cells with KRT5-CreER alleles.

The suggestion that LUSC originates from basal cells comes from a recent study that indicates the gene signature of basal stem cells (BSCs) closely resembles the human lung squamous cell carcinoma (LUSC) gene signature^{8,9}. To test the role of BCL11A in these cells we employed the organoid system whereby basal cells from the mouse were isolated and BCL11A was either overexpressed or knocked out. We demonstrated that BCL11A overexpression in basal organoids lead to abnormal growth with increased markers of proliferation and that its deletion led to loss of organoid formation.

With regard to the Krt5-CreER - Krt5 is ubiquitously expressed in the epidermis of the skin and the basal layer of the mammary gland^{9,10} and BCL11A has been shown to play an important role in the homeostasis of both the mammary epithelia and the epidermis^{11,12} thus, making any long-term lung cancer studies difficult to perform.

5. What is the expression pattern of SETD8 in the mouse model overexpressing BCL11A?

To determine the contribution of BCL11A to the transcriptional regulation of Setd8, we induced BCL11A overexpression using the BCL11A^{ovx} mice and isolated Basal Stem Cells (BASC) and tracheal Epcam⁺ cells (from three independent mice) for qRT-PCR analysis. We found that BCL11A overexpression induced the upregulation of Setd8 by 3 fold (Figure 4h). Therefore, this data indicates that BCL11A plays a direct role in regulating Setd8. Moreover, we probed for SETD8 protein in the mouse model of BCL11A^{ovx} and found an increase in SETD8 in the airways and hyperplastic regions.

Minor: High SOX2 levels do not necessarily mean squamous differentiation. Supplementary Figure 3b). The normal proximal airways (e.g. trachea) have high levels of SOX2.

We thank the reviewer for the comment we have now changed the sentence to read that Sox2 expression is suggestive of squamous differentiation which is supported by

a recent TCGA pan-squamous cancer genomics study SOX2 has been highlighted as a common marker regardless of the tissues subtype¹³

References

1. Zhang, Z. & Guo, Y. *High TBX2 expression predicts poor prognosis in non-small cell lung cancer*, (2014).
2. Hagerstrand, D., *et al.* Systematic interrogation of 3q26 identifies TLOC1 and SKIL as cancer drivers. *Cancer discovery* **3**, 1044-1057 (2013).
3. Chen, T., *et al.* miR-382 inhibits tumor progression by targeting SETD8 in non-small cell lung cancer. *Biomedicine & Pharmacotherapy* **86**, 248-253 (2017).
4. Saiki, Y., Yamazaki, Y., Yoshida, M., Katoh, O. & Nakamura, T. Human EVI9, a Homologue of the Mouse Myeloid Leukemia Gene, Is Expressed in the Hematopoietic Progenitors and Down-Regulated during Myeloid Differentiation of HL60 Cells. *Genomics* **70**, 387-391 (2000).
5. Sankaran, V.G., *et al.* Human Fetal Hemoglobin Expression Is Regulated by the Developmental Stage-Specific Repressor BCL11A. *Science* **322**, 1839 (2008).
6. Xu, J., *et al.* Corepressor-dependent silencing of fetal hemoglobin expression by BCL11A. *Proceedings of the National Academy of Sciences* **110**, 6518 (2013).
7. Kadoch, C., *et al.* Proteomic and bioinformatic analysis of mammalian SWI/SNF complexes identifies extensive roles in human malignancy. *Nature Genetics* **45**, 592 (2013).
8. Weeden, C.E., *et al.* Lung Basal Stem Cells Rapidly Repair DNA Damage Using the Error-Prone Nonhomologous End-Joining Pathway. *PLOS Biology* **15**, e2000731 (2017).
9. Van Keymeulen, A., *et al.* Distinct stem cells contribute to mammary gland development and maintenance. *Nature* **479**, 189 (2011).
10. Blanpain, C. & Fuchs, E. Epidermal Stem Cells of the Skin. *Annual review of cell and developmental biology* **22**, 339-373 (2006).
11. Khaled, W.T., *et al.* BCL11A is a triple-negative breast cancer gene with critical functions in stem and progenitor cells. **6**, 5987 (2015).
12. Li, S., *et al.* Transcription Factor CTIP1/ BCL11A Regulates Epidermal Differentiation and Lipid Metabolism During Skin Development. *Scientific Reports* **7**, 13427 (2017).
13. Campbell, J.D., *et al.* Genomic, Pathway Network, and Immunologic Features Distinguishing Squamous Carcinomas. *Cell Reports* **23**, 194-212.e196 (2018).

Reviewers' comments:

Reviewer #1 (Remarks to the Author):

The authors have adequately responded to the comments of the referee and revised the manuscript accordingly. I support publication.

Reviewer #2 (Remarks to the Author):

The authors have performed extensive experiments to address this reviewer's concerns. The manuscript has also been revised extensively. That said, E-cad staining in the rebuttal letter does not seem to work. The epithelium in the airway should be stained, not the blood vessels next to the airways.

Reviewers' comments:

Reviewer #1 (Remarks to the Author):

The authors have adequately responded to the comments of the referee and revised the manuscript accordingly. I support publication.

We thank the reviewer for the suggestions and comments which improved our manuscript.

Reviewer #2 (Remarks to the Author):

The authors have performed extensive experiments to address this reviewer's concerns. The manuscript has also been revised extensively.

We thank the reviewer for the suggestions and comments which improved our manuscript.

That said, E-cad staining in the rebuttal letter does not seem to work. The epithelium in the airway should be stained, not the blood vessels next to the airways.

We have now repeated the staining with a different antibody which has been shown to work on mouse lung tissue (Laresgoiti U et al. 2016) and indeed the airways are E-CAD positive in Control and BCL11A^{ovx} mice (including the nodules) (see below).